# Characterisation of GFAP-Expressing Glial Cells in the Dorsal Root Ganglion after Spared Nerve Injury

**DOI:** 10.3390/ijms242115559

**Published:** 2023-10-25

**Authors:** Elena A. Konnova, Alexandru-Florian Deftu, Paul Chu Sin Chung, Marie Pertin, Guylène Kirschmann, Isabelle Decosterd, Marc R. Suter

**Affiliations:** 1Pain Center, Department of Anesthesiology, Lausanne University Hospital (CHUV), 1005 Lausanne, Switzerland; 2Department of Fundamental Neurosciences, Faculty of Biology and Medicine, University of Lausanne, 1005 Lausanne, Switzerland

**Keywords:** GFAP, satellite glial cells, non-myelinating Schwann cells, nerve injury, dorsal root ganglion, neuropathic pain

## Abstract

Satellite glial cells (SGCs), enveloping primary sensory neurons’ somas in the dorsal root ganglion (DRG), contribute to neuropathic pain upon nerve injury. Glial fibrillary acidic protein (GFAP) serves as an SGC activation marker, though its DRG satellite cell specificity is debated. We employed the hGFAP-CFP transgenic mouse line, designed for astrocyte studies, to explore its expression within the peripheral nervous system (PNS) after spared nerve injury (SNI). We used diverse immunostaining techniques, Western blot analysis, and electrophysiology to evaluate GFAP+ cell changes. Post-SNI, GFAP+ cell numbers increased without proliferation, and were found near injured ATF3+ neurons. GFAP+ FABP7+ SGCs increased, yet 75.5% of DRG GFAP+ cells lacked FABP7 expression. This suggests a significant subset of GFAP+ cells are non-myelinating Schwann cells (nmSC), indicated by their presence in the dorsal root but not in the ventral root which lacks unmyelinated fibres. Additionally, patch clamp recordings from GFAP+ FABP7−cells lacked SGC-specific K_ir_4.1 currents, instead displaying outward K_v_ currents expressing K_v_1.1 and K_v_1.6 channels specific to nmSCs. In conclusion, this study demonstrates increased GFAP expression in two DRG glial cell subpopulations post-SNI: GFAP+ FABP7+ SGCs and GFAP+ FABP7− nmSCs, shedding light on GFAP’s specificity as an SGC marker after SNI.

## 1. Introduction

Glial cells are involved in neuropathic pain conditions in the peripheral (PNS) and central nervous systems (CNS) [1,2]. Physiologically, they play supportive, protective and modulatory roles. After peripheral nerve injury, they become activated and contribute to the establishment and maintenance of neuropathic pain by sensitising primary neurons at the site of injury, along the nerve and in the dorsal root ganglion (DRG) [2,3].

The DRG contains the soma of primary sensory neurons that connect to the pseudo-unipolar axon via a T-shaped junction [4]. The axons and somas are anatomically segregated into soma-rich and axon-rich regions within the DRG. Glial cells that surround the neuronal somas in the DRG have been collectively referred to as satellite glial cells (SGCs). In the tissue, SGCs are often identified by their morphology, as they distinctively form crescents or rings around the soma of primary sensory neurons [5]. However, other non-neuronal cell types are present in between the neuronal somas, such as immune cells and fibroblasts. For example, macrophages can be found in close proximity to the neuronal soma after nerve injury, but are distinguishable by their specific expression of Iba1, among other markers [6].

A large number of markers have been used to label SGCs such as S100β, sox10, vimentin, PLP1, but they are also expressed in other peripheral glial cell types due to their shared ontogeny [7,8]. SGCs and Schwann cells originate from the glial precursors of the neural crest during development and maintain shared expression of markers [9]. Mature Schwann cells that myelinate large and medium-sized Aβ and Aδ afferent sensory fibres are easily distinguished by the expression of myelin-related proteins, such as myelin basic protein (MBP), myelin protein zero (MPZ) and non-compact myelin associated protein (NCMAP). Non-myelinating Schwann cells (nmSCs), also called Remak cells, envelop the thinner C-fibres without myelinating them. They are often identified in the nerve by the expression of glial markers and lack of myelin markers. Glial fibrillary acidic protein (GFAP), neurotrophin receptor p75NTR and L1 cell adhesion molecule (L1CAM) are examples of markers that are thought to be specific to nmSCs in the nerve in naïve adult rodents [10,11]. Those markers are also expressed in immature Schwann cells during the development and “repair” of Schwann cells after nerve injury [10,11]. GFAP and p75NTR are also expressed in the DRG, but it is unclear which type of glial cells they label: SGCs surrounding the neuronal soma, nmSCs along unmyelinated axons, or perhaps a new glial cell type [4,5,12].

Single-cell RNA sequencing studies determined that fatty acid binding protein 7 (FABP7) is a specific marker of SGCs in the PNS of adult mice and it has been reliably used to distinguish them from other types of peripheral glial cells by immunohistochemistry [13]. However, the classification of glial cells within the DRG is likely more complex than previously defined. Recent transcriptomic data identified subpopulations of SGCs within the DRG, sympathetic ganglion and trigeminal ganglion in naïve mice [8,14]. There may be further diversity generated after nerve injury, as SGCs become activated and undergo expression changes [6]. In the peripheral terminals of Aβ-primary sensory neurons is another type of non-myelinating Schwann-like cells called terminal glial cells that primarily work as low-threshold mechanoreceptors and potentially exhibit neurotransmission properties during action potential generation [15].

After nerve injury, SGCs increase coupling with neighbouring SGCs by upregulating connexin 43 (Cx43) gap junctions and form a network of SGCs that surround multiple neurons [16,17]. Moreover, SGCs may bridge separate perineuronal sheaths, forming complex sheath structures, which exhibit complex neuron-SGC boundaries [18]. Additionally, SGCs mediate cross-sensitisation by electrical coupling to the neuron they surround [19]. SGCs are characterised by large currents from inward rectifying potassium channels K_ir_4.1, which is specific to SGC in the DRG and decreases in various types of neuropathic pain models [8,16,20,21,22,23,24]. Upregulation of GFAP is a commonly used marker of SGC activation [5,25,26]. It is virtually absent from the DRG of naïve rats and mice, but becomes significantly upregulated in neuropathic pain models, such as type-1 diabetes, chemotherapy, and inflammatory pain models. GFAP upregulation in SGCs is prominent in rat models of nerve-injury, but some studies do not observe a significant upregulation in mouse models of nerve-injury [5,6].

SGC have been compared to astrocytes of the CNS because they share similar functions and markers, such as FABP7, GS, S100, K_ir_4.1, Cx43, and they upregulate GFAP upon activation [26,27]. Due to these similarities, several transgenic mouse lines targeting astrocytes were investigated as potential tools to target SGCs in the DRG. Transgenic mouse lines, such as S100β-eGFP, GLAST-CreERT2::GCaMP6f and GFAP-Cre::GCaMP6f, were found not to target SGCs efficiently [28]. Surprisingly, the GFAP-Cre::GCaMP6f mice targeted mainly neurons rather than glial cells. In that study, Cx43-CreERT2::GCaMP6f was the most suitable transgenic line for the purpose of calcium imaging of SGCs [28]. In another study, the FABP7-creER-GCaMP6s mouse line was shown to be suitable for calcium imaging of SGCs [29]. Exploring additional transgenic mouse lines to investigate SGCs remains necessary to better characterise different populations of glial cells in the DRG [30].

In this study, we use the hGFAP-CFP mouse line that was previously developed to label astrocytes in the CNS [31,32] to characterise GFAP-expressing cells in the DRG after nerve injury. Via immunohistochemistry, antibodies against GFAP label the filamentous protein in the processes within the glial cells, which makes the identification of the cell body difficult. A reporter mouse line, such as hGFAP-CFP, expresses the fluorophore in the cytosol, which clearly labels the cell body. Furthermore, it allows us to visualise the GFAP-expressing cells in culture for electrophysiological recording by a patch clamp. Using the hGFAP-CFP mouse line, we aim to characterise the identity of the GFAP-expressing cells in the DRG and the changes ensuing nerve injury.

## 2. Results

### 2.1. Increase in GFAP-Expressing Cells in the DRG after Spared Nerve Injury (SNI) without Proliferation

The hGFAP-CFP mouse line was previously used to label astrocytes in the CNS [32]. We observed CFP expression in the PNS. The CFP signal was present in non-neuronal cells in both regions of the DRG containing primarily neuronal somas, and in axon-rich regions. GFAP protein forms the cytoskeletal structure of glial cells and is found located in the cell’s processes [33], while the endogenous CFP signal labels the cell body of non-neuronal cells in the DRG (Figure 1A,B). To confirm that the CFP-labelled cells are GFAP+, we sorted the cells from the DRG by fluorescence-activated cell sorting (FACS) and immunostained them with the GFAP antibody to reveal that 78.4% ± 11.0 S.D., from the total of CFP+ cells, were co-labelled with the GFAP antibody (Figure 1C,D). We confirmed that none of the CFP+ cells were Iba1+ cells in the DRG (Figure 1E,F). 

GFAP is a commonly used marker to study the reactivity of SGCs in neuropathic pain models [5,25,26]. In the neuronal soma-rich regions in the DRG, we found that the number of CFP+ cells doubles by the 2nd day after SNI and remained elevated at 4 and 7 days post-SNI (Figure 2A–D). Interestingly, the increase in CFP+ cells is not accompanied by their proliferation after SNI (Figure 2A–C,F). Ki67 marks the nucleus at all the active proliferation phases in the cell cycle, but it is absent from cells in the resting phase [34]. While the total expression of Ki67 is increased in the DRG after SNI, it is not significantly expressed in CFP+ cells; <1% of CFP+ cells co-localise with Ki67 staining (Figure 2A–C,E,F). We also observed the upregulation of GFAP protein in the L3-L5 DRG, compared to sham mice (Figure 2G). 

The increase in CFP signal occurred in the L3 and L4 DRG, which contained about 21.5% ± 9.9 S.D. and 36.5% ± 7.8 S.D. of activating transcription factor 3 (ATF3+) injured neurons, respectively (Figure 3A–D). The L5 DRG contained mainly uninjured neurons from the spared sural nerve; therefore, it expressed a negligible amount of ATF3 (activating transcription factor 3) in neurons after SNI (6.6% ± 4.5 S.D.). The percentage values of ATF3+ neurons were calculated based on NeuN staining. In addition, CFP+ cells were found more often close to ATF3+ injured neurons, rather than ATF3− uninjured neurons in the L3 and L4 DRG at 2, 4 and 7 days after SNI (Figure 3E).

### 2.2. Some GFAP+ Glial Cells Are a Subpopulation of Satellite Glial Cells

Previous studies reported the use of GFAP as a gliosis marker in the DRG [5,25,26]. To determine if CFP+ cells are activated SGCs, we immunostained the DRG tissue with antibodies against markers described as specific for SGC: FABP7, GS, Cx43 and K_ir_4.1. All these markers labelled crescent- or ring-shaped cells surrounding the soma of DRG neurons, but were absent from axon-rich regions. GS and K_ir_4.1 antibodies stained the processes and the edges of the cell bodies and the Cx43 antibody revealed a dot pattern on the membrane of the cells, while the FABP7 signal was found in the cytosol of the cell body and processes of the SGCs (Figure 4A–F). We observed poor colocalisation of CFP+ signals from the cells’ soma, and a signal from GS, Cx43 and K_ir_4.1 antibodies, which was the brightest in different cellular compartments, preventing reliable quantification (Figure 4D–F). Only 24.5% ± 9.6 S.D. of CFP+ cells in the soma-rich region of the DRG were found to have co-localisation with the FABP7+ signal (Figure 4A–C,H). The proportion of CFP+ FABP7+ cells was not significantly changed after SNI due to the increase in both CFP+ FABP7+ and CFP+ FABP7− cells (Figure 4A–C,G–I). None of the CFP+ cells in the axon-rich region of the DRG were labelled with either of these SGC markers. 

### 2.3. Some CFP+ Cells Are Non-Myelinating Schwann Cells

Many CFP+ cells are found in the axon-rich regions in the DRG and in the sciatic nerve, leading us to hypothesise that some of the CFP+ cells may be GFAP+ nmSC (Figure 5A,B).

The dorsal root contains large Aβ, Aδ fibres myelinated by Schwann cells, and unmyelinated C fibres enveloped by nmSCs. By contrast, the murine L4 ventral roots contain very few unmyelinated fibres [35], and it primarily consists of large myelinated efferent fibres. Due to this composition of fibres, the ventral root would require myelinating Schwann cells, but not nmSCs. CFP+ cells were found in the dorsal root, DRG, spinal nerve and sciatic nerve, but not in the ventral root (Figure 5A,B). Furthermore, the lack of colocalisation of CFP+ cells with MBP indicates that they are not myelinating Schwann cells (Figure 5D). These results suggest that some CFP+ cells are GFAP-expressing nmSCs. Furthermore, we observed an increase in the CFP+ signal in the ipsilateral sciatic nerve at 7 days after SNI compared to the contralateral side (Figure 5B,C).

Since myelinating Schwann cells, nmSC and SGC share a common ontogeny, many markers, such as PLP1, S100b, sox10, are expressed in all these types of peripheral glial cells [7,8]. L1CAM expression is enriched in nmSC, but also expressed in some DRG neuron subpopulations [8,20]; therefore, we were interested in L1CAM as a marker to discriminate between nmSC and SGC. We found evidence of a signal from the L1CAM antibody in the DRG, spinal nerve, dorsal root, but not the ventral root, and that appeared to co-localise with CFP+ cells (Appendix A). However, after DRG dissociation, we find that L1CAM is strongly expressed in the soma and axon of some neurons, and less brightly in some CFP+ cells and many L1CAM− CFP+ cells are found along the L1CAM+ neuronal axons (Appendix A). 

### 2.4. Electrophysiological Properties of CFP+ FABP7− Cells

We performed whole-cell patch clamp recordings on CFP+ cells from DRG dissociated cultures to further characterise them at 7 days after SNI. We targeted CFP+ cells that were detached from neuronal somas. Staining the dissociated DRG culture after fixation with the FABP7 antibody revealed that FABP7+ CFP+ cells remain attached to the neuronal soma, but all detached CFP+ cells were FABP7− (Figure 6A). 

Voltage clamp recordings show that the detached CFP+ cells have no inward potassium currents that could be attributed to K_ir_4.1 channels (Figure 6B). K_ir_4.1 is known to be specifically expressed in FABP7+ SGC in the DRG [7,8,20]. Instead, we find that detached CFP+ cells in the DRG culture have large outward currents that are partially inhibited by voltage-gated potassium channel (K_v_) inhibitor 4AP (4-Aminopyridine) (Figure 6B). The outward current density and resting membrane potential are similar between CFP+ cells from ipsilateral or contralateral DRG at 2 and 7 days after SNI (Figure 6B–D). Western blot analysis of CFP+ cells sorted by FACS revealed the presence of K_v_1.1 and K_v_1.6 channels and that the expression level is similar between samples from ipsilateral and contralateral DRG at 7 days after SNI (Figure 6F–I). 

## 3. Discussion

In this study, we distinguished two subpopulations of GFAP+ cells within the DRG using the hGFAP-CFP transgenic mouse line: FABP7+ SGCs and FABP7− nmSCs. Recently, FABP7 has been used as the preferred marker to identify SGCs because it was shown to be specifically expressed in SGCs in the PNS of adult mice [8,13]. FABP7 is also a useful marker to identify SGCs in rats and humans [36]. We found that 24.5% of GFAP+ glial cells were FABP7+ SGCs in the soma-rich region of the murine DRG. However, many FABP7+ SGCs did not express GFAP, even after peripheral nerve injury, indicating that GFAP is not a suitable marker to label all SGCs. A low level of GFAP can be found in the SGCs of naïve rats, but it is undetectable in SGCs from mice and humans [5,6,36].

We found that 75.5% of GFAP-expressing glial cells in the soma-rich region of the DRG are FABP7−. We deduced that GFAP+ FABP7− glial cells in the DRG are nmSCs, as they are also present in the axon-rich region of the DRG, in the dorsal root, spinal nerve and sciatic nerve, where unmyelinated C-fibres are present. By contrast, GFAP-expressing cells were absent from the mouse ventral root, which mainly consists of myelinated efferent motor fibres and does not contain unmyelinated fibres [35,37]. Furthermore, GFAP-expressing cells did not co-localise with the myelin protein MBP, indicating that they are not myelinating Schwann cells.

GFAP-expressing nmSC within the soma-rich region of the DRG may line the proximal T-junction axon connecting to the neuronal soma. However, we cannot exclude the presence of other GFAP-expressing glial cell types. Single-cell RNA sequencing studies have identified multiple subpopulations of glial cells in the DRG, sympathetic ganglion and trigeminal ganglion that were labelled as SGCs [8,14]. However, it is unclear which of these subpopulations express GFAP before and/or after nerve injury. These transcriptomic “SGCs” subpopulations may in fact be distinct glial cell types. For example, 75NTR+ GS− GFAP− glial cells in the naïve DRG were identified to be morphologically distinct from SGC and nmSCs, lying on top of SGCs at the level of the proximal segment of the axon close to the soma [4]. Some studies have shown that after peripheral nerve injury, 75NTR+ glial cells express GFAP [38,39,40]. Therefore, we cannot ignore the fact that a portion of GFAP-expressing glial cells in the soma-rich region of the DRG after nerve injury may be 75NTR+ glial cells.

L1CAM is a marker used to identify nmSCs and immature Schwann cells in peripheral nerves [9,41]. We found that the L1CAM marker was not convenient to identify nmSCs in the murine DRG tissue. Immunohistochemistry for L1CAM also marked the primary sensory neurons’ soma and axon. In addition, many L1CAM− GFAP+ glial cells are present along the L1CAM+ neuronal axons. Nevertheless, we did find some L1CAM+ GFAP+ nmSCs in the dissociated culture. Transcriptomic data indicate that L1CAM is expressed in nmSCs but also primary sensory neurons [8,20], particularly in non-peptidergic (NP) and peptidergic (PEP1) nociceptive neurons [20]. Those nociceptive neurons have unmyelinated C-fibres, a small soma in the DRG and are enwrapped by nmSCs [42]. GFAP-expressing glial cells wrapping around L1CAM+ nociceptive neurons may be falsely identified as L1CAM+ nmSCs by immunohistochemistry. There remains a lack of a validated marker to specifically distinguish nmSCs from all other types of neuronal and non-neuronal cells in the DRG. Entpd2 and Scn7a are upregulated genes in nmSCs and may be good candidates [8,43]. However, Entpd2 is expressed at lower levels in SGCs and fibroblasts [43], while Scn7a may be also expressed in a subpopulation of SGCs in the DRG [14].

We also confirmed the identity of GFAP-expressing nmSCs in the DRG by electrophysiological characterisation. GFAP+ FABP7− cells easily detach from neuronal somas following DRG dissociation. Those cells were identified as nmSCs for their lack of the SGC-specific K_ir_4.1 inward currents, and the presence of large K_v_ outward currents. Staining for K_ir_4.1 specifically labels SGCs that surround the neuronal somas and are absent from the axon-rich region of the DRG. Furthermore, many GFAP-expressing glial cells in the soma-rich region of the DRG did not express K_ir_4.1, indicating the presence of other glial cells close to the neuronal soma. 

We show that K_v_1.1 and K_v_1.6 channels are present in GFAP+ glial cells in the DRG. Both channels are strongly expressed in nmSCs [8,20]. Additionally, K_v_1.1 is also expressed in myelinating Schwann cells but absent from SGCs, while K_v_1.6 expression is detectable in SGCs but absent from myelinating Schwann cells [8,20]. In addition to K_v_1.1 and K_v_1.6, K_v_1.2 is also expressed in nmSCs, but it is also expressed at a lower level in SGCs and myelinating Schwann cells [8,20]. All three K_v_ channels may contribute to the large outward potassium currents in GFAP-expressing nmSCs that were inhibited by the non-specific K_v_ channel inhibitor 4AP. These results support the existence of GFAP+ nmSCs in the DRG. 

To our knowledge, this is the first study confirming the presence of K_v_ currents in nmSC from adult mice by a patch clamp [44]. These electrophysiology findings are congruent with transcriptomic data of adult nmSCs, demonstrating the expression of K_v_1.1, K_v_1.2 and K_v_1.6, but not K_ir_4.1 [8,20]. Interestingly, in embryonic and post-natal development, nmSCs of the nerve have K_ir_4.1 currents [45,46]. Perhaps, K_ir_4.1 currents become reduced in nmSCs, while they are maintained in SGCs in adulthood during development and maturation. 

After SNI, we observe an increase in GFAP+ glial cells without proliferation, specifically in the L3 and L4 DRG, where injured neurons are present. Furthermore, more GFAP+ cells were located near injured, rather than uninjured neurons within the L3 and L4 DRG. These results suggest neuro-glial cross talk, where signals from injured neurons, such as nitric oxide [47] or ATP [48], could activate glial cells and induce the upregulation of GFAP. Some studies report an increase in the proliferation of SGCs in the DRG tissue after nerve injury [12,49,50], but other studies determined that proliferation occurs mainly in macrophages, rather than SGCs [6,13]. It can be challenging to visually attribute the proliferating nucleus to the correct non-neuronal cell type using markers that label cell processes rather than the cell body, due to the density of the DRG tissue. Here, we cannot fully exclude the proliferation of GFAP− SGCs. 

We found that the post-SNI increase in GFAP-expressing cells is partially due to the upregulation of GFAP in FABP7+ SGCs, indicating their change in phenotype. Besides SGCs, there is also a large increase in GFAP+ FABP7− nmSCs in the soma-rich region of the DRG. GFAP-expressing nmSCs in the DRG may be able to migrate towards the neuronal soma. Moreover, a global re-organisation of non-neuronal cells in the DRG following nerve injury is seen, with previous reports indicating the infiltration of macrophages between the SGCs and neurons [6]. 

Alternatively, the increase in GFAP+ FABP7− cells may stem from the appearance of repair Schwann cells. In nerves, the Wallerian degeneration leads to the de-myelination of axons and reprogramming of myelinating Schwann cells and nmSCs into repair Schwann cells [11,51,52,53]. Repair Schwann cells undergo reprogramming that downregulates the expression of myelin proteins and upregulates markers of nmSCs, such as GFAP, L1CAM and p75NTR, along with de novo expression of genes Shh and Olig1 [51,54]. We also observed an upregulation of GFAP expression in the sciatic nerve after SNI. A similar process of reprogramming of myelinating Schwann cells and nmSCs into repair Schwann cells may be occurring at the level of the DRG.

We found no significant changes in K_v_ outward currents in GFAP+ FABP7− nmSCs at 2 and 7 days after SNI. Although we did not find changes in the electrophysiological state of nmSCs within the DRG, functional changes may still occur that could impact the repair and sensitisation of primary sensory neurons after SNI [13,52,55]. A better understanding of glial subpopulations present within the DRG will set the path for the investigation of the differential role of each glial cell type following peripheral nerve injury.

The study’s findings have translational implications for understanding the diverse glial cell subpopulations within the DRG in the context of peripheral nerve injury. The identification of distinct subtypes, such as FABP7+ SGCs and FABP7− nmSCs, underscores the complexity of the DRG microenvironment. These insights may lead to the development of targeted therapies for neuropathic pain management by focusing on specific glial cell subtypes. However, challenges in accurately distinguishing these subpopulations due to the lack of definitive markers emphasize the need for further research.

Several limitations warrant consideration in this study. While our research identified distinct GFAP+ cell subpopulations within the DRG, it primarily relied on the hGFAP-CFP transgenic mouse line, potentially limiting its generalisability to other species or models. The use of FABP7 as a marker for SGCs raises questions about its universal applicability, especially since not all FABP7+ SGCs expressed GFAP, suggesting that GFAP may not be all-encompassing for SGC labelling. Additionally, the study’s focus on the soma-rich region of the murine DRG may have overlooked potential variations in other regions. The lack of a definitive marker for nmSCs highlights the ongoing challenge in distinguishing these cells from other DRG constituents. Finally, while electrophysiological characterisation offered valuable insights, it did not reveal significant functional changes in nmSCs post-injury, which may be revealed using other techniques. Clarifying the specific roles of diverse glial cell subpopulations within the DRG remains a complex, evolving endeavour. 

## 4. Materials and Methods

### 4.1. Animals 

The experiments were performed on homozygote TgN(hGFAP-ECFP)-GCED referred to as hGFAP-CFP mice (a generous gift from Paola Bezzi, Department of Fundamental Neurosciences, University of Lausanne), including adult male and female mice [32]. The animals were housed in standard cages with free access to food and water at 22 ± 0.5 °C under a controlled 12 h light/dark cycle. The SNI or sham surgeries were performed unilaterally as described previously [56,57]. Briefly, the mice were anesthetised with isoflurane (Piramal, Mumbai, Maharashtra, India). The trifurcation of the sciatic nerve was exposed. The two branches of the nerve (tibial and common peroneal) were ligated with silk 6.0 sutures and sectioned, while the sural branch remained spared. The muscles and skin were sutured and animals were left to recover before the transfer into standard cages. Tissue was collected after terminal intraperitoneal injections of pentobarbital (50 mg/kg) (Streuli Pharma, Uznach, Switzerland). Detailed references for the resources used are presented in Appendix A.

### 4.2. Dissociation and FACS

Ipsilateral and contralateral L3–L5 DRG were surgically removed and immersed in cold Hanks’ Balanced Salt Solution (HBSS, Sigma-Aldrich, Saint-Louis, MO, USA). Subsequently, DRG were incubated for 1 h at 37 °C with 1 mg/mL dispase II (Roche, Basel, Switzerland) and 2 mg/mL collagenase A (Roche, Basel, Switzerland) prepared in HBSS. Using a fire polished glass Pasteur pipette, DRG were transferred to 1 mL of warm culture medium: Dulbecco’s Modified Eagle Medium (DMEM, Gibco, Billings, MT, USA) supplemented with 10% foetal bovine serum (FBS) (Gibco, Billings, MT, USA) and 1% penicillin/streptomycin (Sigma-Aldrich, Saint-Louis, MO, USA). DRG were mechanically dissociated by triturating 10–13 times. Then, 4 mL of culture medium was added to dilute the enzymes and the dissociated DRG were centrifuged at 1000 rpm for 10 min. The supernatant was discarded and the DRG pellet was re-suspended in the appropriate volume depending on the subsequent technical approach. At the end, 50 μL of re-suspended cells was plated on a poly-D-lysine (Sigma-Aldrich Inc., Saint-Louis, MO, USA) coated coverslip, or in 500 µL for subsequent sorting by FACS. Samples were filtered through a cell strainer Snap-Cap with a 35 um filtering mesh into falcon test tubes (Corning, Corning, NY, USA) for FACS of CFP+ cells with a 488 mm laser, at room temperature (21–23 °C) using MoFlo Astrios EQ (Beckman Coulter, Brea, CA, USA). Positive events were harvested in culture medium for subsequent in vitro culture or Western blot analyses. Detailed references for the resources used are presented in Appendix A.

### 4.3. Western Blot

Western blot analyses were performed on DRG tissue or CFP+ cells sorted by FACS. The samples were lysed in 100 mM Tris HCl pH 6.8, 2% SDS, 10% glycerol, Complete Protease inhibitor cocktail tablets (Roche, Basel, Switzerland). Soluble fractions were recovered in supernatants after 20 min of centrifugation at 10,000 rpm at 4 °C. Protein concentration was measured using the Bradford BSA standard scale (Sigma-Aldrich, Saint-Louis, MO, USA) with Assay dye (BioRad, Hercules, CA, USA) and loaded at an amount of 40 µg/well. For the Western blot analysis of GFAP protein in L3, L4, L5 DRGs tissue, normalisation was performed to tubulin, whereas the quantification of the K_v_1.1 and K_v_1.6 protein level in CFP+ cells sorted by FACS was normalised to the total protein level. Proteins were separated on acrylamide SDS-PAGE and then transferred to polyvinylidene fluoride membranes (BioRad, Hercules, CA, USA) that were immunoblotted with the following antibodies: rabbit anti-GFAP (1:100 in milk, Aligent, Santa Clara, CA, USA), rabbit anti- K_v_1.1 (Thermo Fisher Scientific, Waltham, MA, USA), rabbit anti- K_v_1.6 (Thermo Fisher Scientific, Waltham, MA, USA) and mouse anti-α-tubulin used as a reference protein (1:20,000 in milk; Sigma-Aldrich, Saint-Louis, MO, USA). We used secondary HRP-linked goat anti-rabbit (1:10,000 in milk, Aligent, Santa Clara, CA, USA) or HRP-linked goat anti-mouse antibodies (1:10,000 in milk, Aligent, Santa Clara, CA, USA), and SuperSignal West Dura Extended Duration Substrate (Thermo Fisher Scientific, Waltham, MA, USA) for detection. Chemiluminescence was detected by using an imaging system (LAS-4000 Imaging System, GE Healthcare, Chicago, IL, USA) coupled with an integrated CCD camera. Protein quantification was performed using ImageJ software https://imagej.net/ij/index.html (Fiji) [58], analysing the ratio between the intensity of the protein band against the tubulin (for DRG tissue) or the total amount of protein (for CFP+ cells sorted by FACS). The quantified signals for the proteins of interest were normalised to reference protein signals. Detailed references for the resources used are presented in Appendix A.

### 4.4. Immunofluorescence

Mice were terminally anesthetised by i.p injection of pentobarbital and transcardially perfused with saline, followed by 4% paraformaldehyde (PFA, Sigma-Aldrich, Saint-Louis, MO, USA) in PBS. L3–L5 DRGs were dissected and post-fixed at 4 °C for 90 min and then transferred in 20% sucrose in PBS for 24 h. The DRG were rapidly frozen in embedding solution (Tissue-Tek O.C.T. Compound, Sakura Finetek, Alphen aan den Rijn, Netherlands) to be cut in 12 μm thickness sections with a cryostat, placed directly onto slides for immunostaining. Alternatively, immunostainings were carried out on dissociated in vitro cultures after overnight incubation on coverslips that were previously coated with poly-D-lysine (Sigma-Aldrich Inc., Saint-Louis, MO, USA). The coverslips were incubated with 4% PFA in PBS for 2 h to fix the cells. Following PBS rinsing, the coverslips or cryosections were immersed in a blocking solution, comprising 10% normal goat serum (NGS, Vector Laboratories, Newark, CA, USA) or normal horse serum (NHS, Vector Laboratories, Newark, CA, USA), and 0.3% Triton X-100 (Sigma-Aldrich, Saint-Louis, MO, USA) in PBS, for 30 min at room temperature. The following primary antibodies were diluted in 5% NGS or NHS in 0.1% triton X-100, and incubated overnight at 4 °C: rabbit anti-Iba1 at 1:500 (Wako Pure Chemical Industries Ltd., Richmond, VA, USA), rabbit anti-GFAP at 1:1000 (Aligent, Santa Clara, CA, USA), rabbit anti-Ki67 at 1:500 (Merck Millipore, Burlington, MA, USA), rabbit anti-GS (1:1000, Abcam, Waltham, MA, USA), rabbit anti-Fabp7 (1:1000 in milk, Thermo Fisher Scientific, Waltham, MA, USA), rabbit anti-connexin43 (1:100, Cell Signalling Technology, Danvers, MA, USA), guinea pig anti- K_ir_4.1 (1:1000, Thermo Fisher Scientific, Waltham, MA, USA), rabbit anti-L1CAM (1:500 in milk, Merck Millipore, Burlington, MA, USA), rabbit anti-ATF3 (1:500, Abcam, Waltham, MA, USA) and chicken anti-NeuN (1:500, Neuromics, Edina, MN, USA). After 3 washes in PBS, the sections were incubated in donkey Cy3 anti-rabbit (1:400, Jackson ImmunoResearch, Ely, UK), goat Alexa Fluor 594 anti-guinea pig (1:1000, Molecular Probes, Eugene, OR, USA), goat Alexa Fluor 594 anti-chicken (1:200, Molecular Probes, Eugene, OR, USA), donkey Cy3 anti-goat (1:300, Jackson ImmunoResearch, Ely, UK), or goat Alexa Fluor 350 anti-rabbit (1:200, Molecular Probes, Eugene, OR, USA) secondary antibodies diluted in 1% NGS or 1% NHS in 0.1% triton X-100 for 2 h at RT. After three final washes in PBS, the sections were dried and covers were mounted with Mowiol 4–88 medium (Calbiochem, San Diego, CA, USA). Epifluorescence images of the DRG sections were acquired at ×20 magnification with the Zeiss Axio Scan.Z1 slide scanner using Zeiss 3.1 Blue software (Zeiss, Oberkochen, Germany) or at ×20 or ×63 with AxioVision (Zeiss, Oberkochen, Germany). Fluorescence intensity and exposure time were kept constant for all images. Cell counting was performed using the Cell Counter Plugin in ImageJ software (Fiji) [58]. Detailed references for the resources used are presented in Appendix A.

### 4.5. Whole-Cell Patch Clamp

L3-L5 DRG cultures were prepared from hGFAP-CFP mice at 2 days and 7 days after SNI. Cells from dissociated DRG were plated on coverslips that were previously coated with poly-D-lysine (Sigma-Aldrich Inc., Saint-Louis, MO, USA). After overnight incubation, the isolated hGFAP-CFP+ cells (not attached to a neuronal soma) in the culture were clamped at −80 mV in whole-cell patch clamps. The potassium currents were recorded using a voltage protocol consisting of a −80 mV hold followed by increasing steps of 10 mV for 300 ms from −140 mV to 100 mV. Sustained currents were measured and normalised to the capacitance of each cell to report the current density (pA/pF) as I-V curves. Subsequently, the recordings were made on the same cell in the current clamp to record the resting membrane potential of the cell. 

Data were acquired using a MultiClamp 700B amplifier and a Digidata 1440A driven by the pClamp 10.3 software (Molecular Devices, San Jose, CA, USA). All whole-cell patch-clamp recordings were made using a BX51WI fixed-stage upright microscope (Olympus, Tokyo, Japan) with a CoolLED pE-340fura equipped with an LED eGFP pE-300 filter-set (CoolLED Ltd., Andover, UK) and an ORCAFlash2.8 digital camera (Hamamatsu Photonics) under the control of the CellSens v3.2 acquisition software (Olympus, Tokyo, Japan). Borosilicate patch pipettes with filaments (Sutter Instruments, Novato, CA, USA) were pulled using a P-97 pipette puller and fire-polished using an MF200-2 microforge, H4 platinum/iridium wire (World Precision Instruments, Sarasota, FL, USA) and W30S-LED Revelation III (LW Scientific, Lawrenceville, GA, USA) at a resistance near 5 MΩ for GFAP+ cells. The fast, slow and whole-cell compensations were made with Multiclamp 700B (Molecular Devices, San Jose, CA, USA) at a bandwidth of 5 kHz and a low-pass filter of 10 kHz.

The extracellular solution to record currents of the GFAP-CFP+ cells contained (in mM) 120 NaCl, 20 KCl, 2 CaCl2, 1 MgCl2, 10 HEPES, 10 D-glucose, and the pH was adjusted to 7.4 with NaOH and the osmolarity was set to between 305 and 310 mOsm with glucose. The intracellular solution contained (in mM) 5 NaCl, 130 KCl, 1 CaCl2, 2 MgCl2, 10 HEPES, 10 EGTA, and the pH was adjusted to 7.35 with KOH and the osmolarity was set between 290 and 300 mOsm with glucose. The intracellular solution was filtered using Nalgene 4 mm syringe filters (176-0020, Thermo Fisher Scientific, Waltham, MA, USA). A SevenCompact™ S210 instrument (Mettler Toledo, Columbus, OH, USA) was used for adjusting the pH of the solutions and an Osmometer 3320 (Advanced Instruments Inc., Norwood, MA, USA) was used to calibrate the osmolarity. Detailed references for the resources used are presented in Appendix A.

### 4.6. Statistics

Statistical analyses were carried out on GraphPad Prism 10 (GraphPad Software, Boston, MA, USA). Two-way ANOVA was performed on immunohistochemistry cell counting and voltage clamp data to analyse the relationship between two variables. Post-hoc multiple comparisons analyses were performed when the intersections of the two variables were significantly different. One-way ANOVA was performed on immunostaining cell counting to account for one variable. When 2 groups were compared, the data were analysed using an unpaired 2-tailed Student’s *t*-test. The N values are mice replicates for immunohistochemistry quantification, cell replicates for patch clamp experiments, and sample replicates for Western blots. Data are represented as mean ± S.D. with individual values (individual dots). *p* values < 0.05 were considered significant results. Thresholds were represented as follows: * *p* < 0.05, ** *p* < 0.01, *** *p* < 0.001, **** *p* < 0.0001. The statistical tests, t, F, and *p* values are provided in Appendix A.

### 4.7. Study Approval

All experiments involving animals were approved by the Committee on Animal Experimentation of the Canton de Vaud, Switzerland, with the experiment licenses VD3068.2b and VD1339.10, in accordance with the Swiss Federal Laws on Animal Welfare and the guidelines of the International Association for the Study of Pain guidelines for the use of animal in research [59].

## 5. Conclusions

This study describes two distinct GFAP+ cell subpopulations within the DRG using the hGFAP-CFP mouse model: FABP7+ SGCs and FABP7− nmSCs. While FABP7 has been preferred as a marker for SGCs, our findings reveal that GFAP is neither a specific nor a sensitive SGC marker. Indeed, 24.5% of GFAP+ glial cells were identified as FABP7+ SGCs in the soma-rich DRG region and a notable fraction of FABP7+ SGCs did not express GFAP, even after nerve injury. The 75.5% of GFAP-expressing glial cells in the soma-rich DRG that lacked FABP7 are thought to be nmSCs. Electrophysiological characterisation confirmed the presence of GFAP+ nmSCs, displaying large K_v_ outward currents, and the expression of K_v_1.1 and K_v_1.6 channels typical of nmSCs, but not kir4.1 currents characteristic of SGC. Although the study did not detect significant changes in K_v_ outward currents, the presence of GFAP+ nmSCs may still impact primary sensory neuron repair and sensitisation following nerve injury. This insight will facilitate further investigations into the distinct roles of glial cell subpopulations in DRG following peripheral nerve injuries.

## Figures and Tables

**Figure 1 ijms-24-15559-f001:**
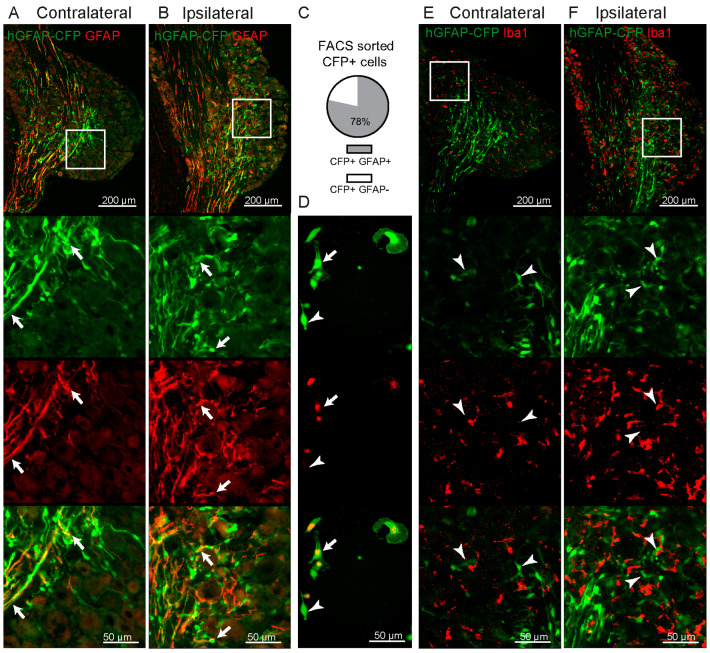
Validation of CFP expression in the DRG of hGFAP-CFP mice. Representative images of immunofluorescence staining with antibody against GFAP of contralateral (**A**) and ipsilateral (**B**) L4 DRG 7 days after SNI. Top panels show the anatomical structure of the DRG, followed by zoomed images (white box) of the endogenous CFP signal (green), antibody staining (red) and merge, with arrows pointing to co-localisation. CFP+ FACS sorted cells after dissociation of ipsilateral L3-L5 DRG at 7 days after SNI were cultured overnight and immunostained with antibody against GFAP (red) to allow for clear quantification of co-labelling of CFP+ GFAP+ cells (green, N = 6 DRG cultures) with arrows pointing to co-localisation and arrowheads to CFP+ GFAP− cells (**C**,**D**). Representative images of immunofluorescence staining with antibody against macrophage marker Iba1 of contralateral (**E**) and ipsilateral (**F**) L4 DRG 7 days after SNI, followed by zoomed image (white box) of the endogenous CFP signal (green), antibody staining (red) and merging, with arrowheads pointing to CFP+ Iba1− cells.

**Figure 2 ijms-24-15559-f002:**
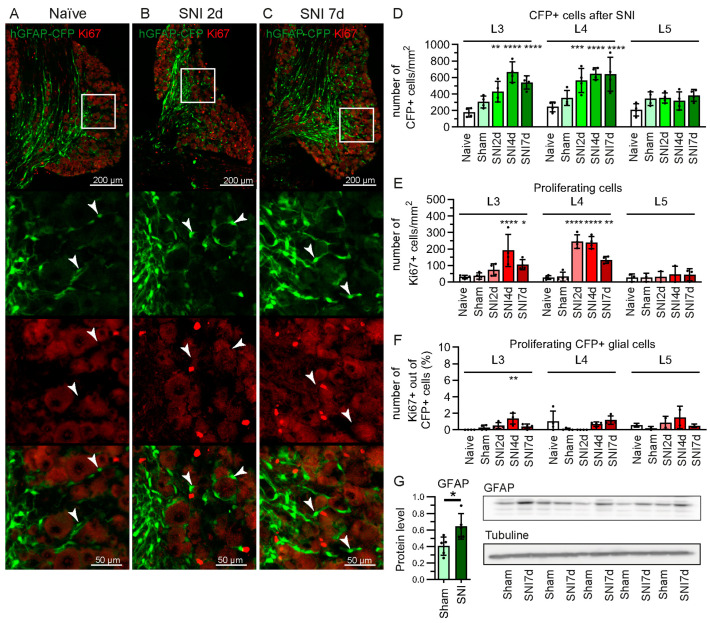
Increase in CFP+ cells in hGFAP-CFP mice without proliferation. Representative images of immunofluorescence staining against proliferation marker Ki67 of L4 DRG from naïve mice (**A**), and at 2 (**B**) and 7 days after SNI (**C**). Top panels show the anatomical structure of the DRG, followed by zoomed image (white box) of the endogenous CFP signal (green), antibody staining (red) and merging, with arrowheads pointing to CFP+ Ki67− cells. Quantification of CFP+ cells (**D**), of Ki67+ cells (**E**), and CFP+ Ki67+ cells (**F**) in the L3, L4 and L5 DRG at 2, 4 and 7 days after SNI, 7 days after sham surgery, and naïve mice (N = 4 mice, two-way ANOVA: D. Interaction F(8, 45) = 3.012 *p* < 0.01, (**E**) Interaction F(8, 40) = 7.193 *p* < 0.0001, (**F**)Interaction F(8, 35) = 2.264 *p* < 0.05, with post-hoc Dunnett’s multiple comparisons to naïve group). (**G**) Western blot for GFAP protein in L3, L4, L5 ipsilateral DRG from mice 7 days after SNI or sham surgery, normalised to tubulin (N = 5 mice, unpaired Student’s *t*-test). * *p* < 0.05, ** *p* < 0.01, *** *p* < 0.001, **** *p* < 0.0001.

**Figure 3 ijms-24-15559-f003:**
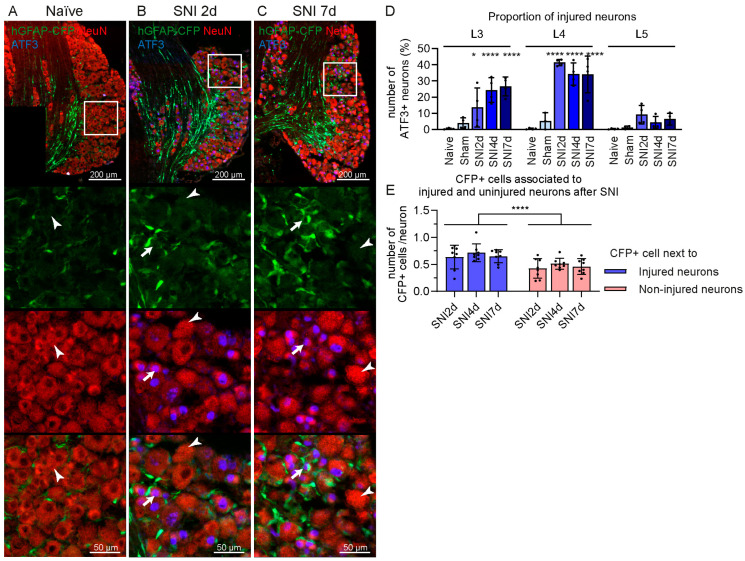
CFP+ cells are preferentially located in proximity of ATF3+ injured neurons in the DRG after SNI in hGFAP-CFP mice. Representative images of immunofluorescence staining against neuronal marker NeuN (red) and injured neurons marker ATF3 (blue) of L4 DRG from naïve mice (**A**), at 2 (**B**) and 7 days after SNI (**C**). Top panels show the anatomical structure of the DRG, followed by zoomed image (white box) of the endogenous CFP signal (green), antibodies staining (red and blue) and merging, with arrows pointing to NeuN+ ATF3+ injured neurons and arrowheads pointing to NeuN+ ATF3− uninjured neurons. (**D**) Quantification of ATF3+ neurons in the L3, L4 and L5 DRG at 2, 4 and 7 days after SNI, 7 days after sham surgery, and naïve mice (N = 4 mice, two-way ANOVA: interaction F(8,43) = 8.411 *p* < 0.0001, post-hoc Dunnett’s multiple comparisons with naïve group). (**E**) Quantification of CFP+ cells in proximity of ATF3+ and/or ATF3− neurons in the L3 and L4 DRG at 2, 4 and 7 days after SNI. (N = 8 DRGs, two-way ANOVA: adjacent neuron type F(1, 42) = 19.17 *p* < 0.0001). * *p* < 0.05, **** *p* < 0.0001.

**Figure 4 ijms-24-15559-f004:**
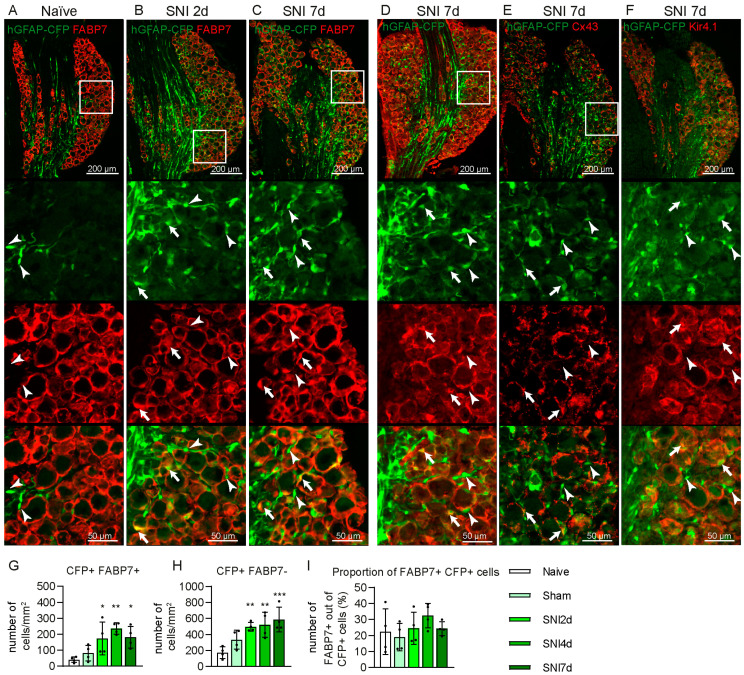
Expression of satellite glial cells (SGCs) markers in CFP+ cells in the DRG after SNI in hGFAP-CFP mice. Representative images of immunofluorescence staining against SGC marker FABP7 of L4 DRG from naïve mice (**A**), at 2 (**B**) and 7 days after SNI (**C**). Representative images of immunofluorescence staining against SGC markers glutamine synthetase (GS) (**D**) and connexin 43 (Cx43) (**E**) and of the K_ir_4.1 channel (**F**) at 7 days after SNI. Top panels show the anatomical structure of the DRG, followed by zoomed image of the endogenous CFP signal (green), antibodies staining (red) and merging, with arrowheads pointing to co-localisation and arrows to CFP+ cells without co-localisation. Quantification of CFP+ FABP7+ (**G**), CFP+ FABP7− cells (**H**) and their proportion (**I**) in the L4 DRG at 2, 4 and 7 days after SNI, 7 days after sham surgery, and naïve mice (N = 4 mice, one-way ANOVA: (**G**) F = 6.671 *p* < 0.01, (H). F = 7.825 *p* < 0.01 (I). F = 1.068 *p* > 0.05, with post-hoc Dunnett’s multiple comparisons with naïve group). * *p* < 0.05, ** *p* < 0.01, *** *p* < 0.001.

**Figure 5 ijms-24-15559-f005:**
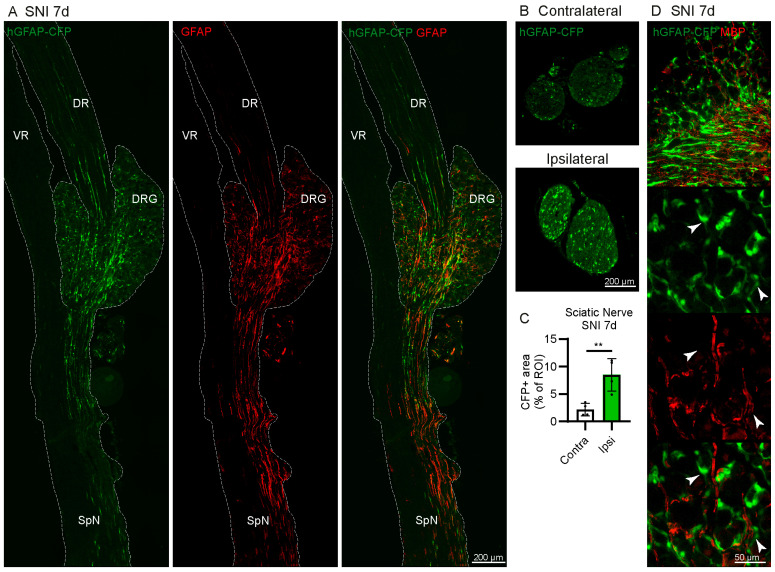
CFP is expressed in non-myelinating Schwann cells (nmSC) in the peripheral nervous system of hGFAP-CFP mice. (**A**) Representative images of immunofluorescence staining against GFAP in L4 DRG with ventral root (VR), dorsal root (DR) and spinal nerve (SpN) at 7 days after SNI. The tissue is outlined by a dotted line for clarity. (**B**) Representative image of endogenous CFP+ signal in transverse section of the contralateral and ipsilateral sciatic nerve 7 days after SNI. (**C**) Quantification of the area of CFP+ signal over the entire region of interest (ROI) manually drawn around the perimeter of the transverse section of the nerve (N = 4 mice, paired Student’s *t*-test). (**D**) Representative images of immunofluorescence staining against myelinating Schwann cells marker myelin basic protein (MBP) at 7 days after SNI. Top panels show the anatomical structure of the DRG with the outlined region of interest containing neuronal somas, followed by zoomed image, with arrowheads pointing to CFP+ MBP− cells. ** *p* < 0.01.

**Figure 6 ijms-24-15559-f006:**
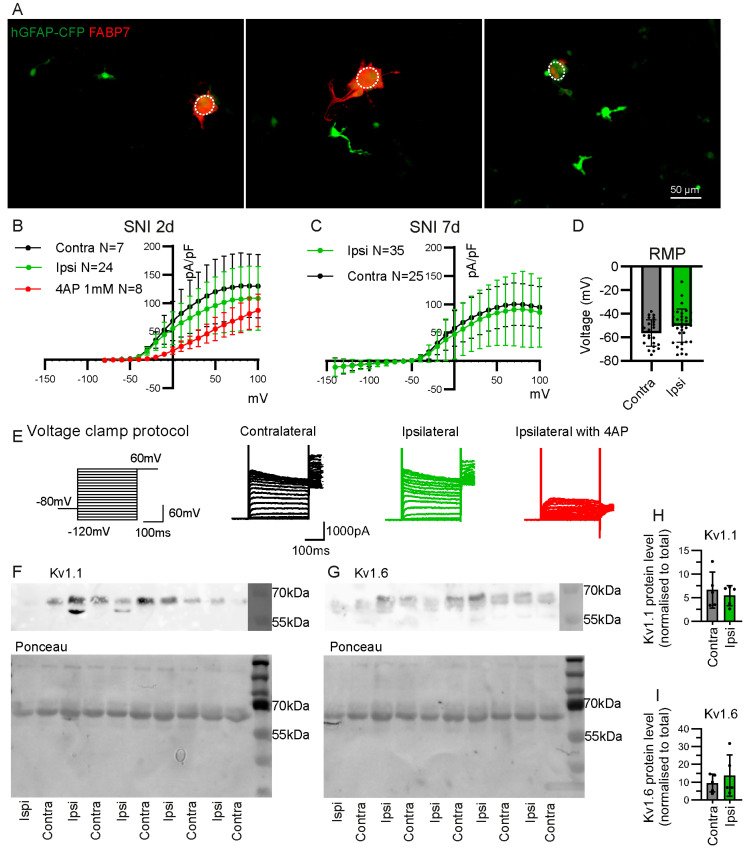
Electrophysiological characteristics of CFP+ FABP7− cells from the DRG of hGFAP-CFP mice. (**A**) Representative images of dissociated DRG culture from hGFAP-CFP (green) mice at 7 days after SNI, with immunofluorescence staining against SGC marker FABP7 (red). Dotted circle delineates the neuronal soma. All observed CFP+ cells that are detached from neuronal soma are FABP7−. Potassium current density of detached CFP+ cells in dissociated culture of contralateral or ipsilateral L3, L4 and L5 DRG, at 2 days (**B**) and 7 days (**C**) after SNI. The 4AP at 1 mM was used to block K_v_ currents (two-way ANOVA: B. Interaction Group F(2, 722) = 48.37 *p* < 0.0001 with post-hoc Dunnett’s multiple comparisons with ipsilateral group ipsi vs. ipsi with 4AP 0 mV to 80 mV *p* < 0.05, (**C**) Interaction Group F(1, 1450) = 8.568 *p* < 0.01 with post-hoc Sidak multiple comparisons contra vs. ipsi −140 mV to 100 mV *p* > 0.05). (**D**) Resting membrane potential (RMP) of the patched detached CFP+ cells at 7 days after SNI (contra N = 25 cells, ipsi N = 35 cells, unpaired Student’s *t*-test *p* > 0.05). (**E**) Voltage clamp protocol and representative traces recorded in whole-cell patch clamp from contralateral and ipsilateral cells with and without 4AP application. Western blot for K_v_1.1 (**F**), K_v_1.6 (**G**), with respective ponceau staining for total protein in samples of CFP+ cells sorted by FACS from L3, L4, L5 ipsilateral and contralateral DRG, 7 days after SNI. Quantification of K_v_1.1 (**H**) and K_v_1.6 (**I**) protein level normalised to total protein level (N = 5 samples, paired Student’s *t*-test *p* > 0.05).

## Data Availability

The data presented in this study are openly available in Zenodo.org at https://doi.org/10.5281/zenodo.10022032.

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
