# Peer review of "Characterisation of GFAP-Expressing Glial Cells in the Dorsal Root Ganglion after Spared Nerve Injury"

_ijms, 2023, doi:10.3390/ijms242115559_

Round 1

Reviewer 1 Report

Title

Multiple subtypes of GFAP-expressing glial cells in the dorsal root ganglion after nerve injury.

Overall view

The author uses the h GFAP-CFP mice line, distinguishing the two types of GFAP subpopulations of glial cells: GFAP+FABP7+SGCs and GFAP+FABP7-nmSCs. The authors use a lot of different methods to confirm the result, immunostaining, Western blot, patch clam. This manuscript had clean result. The article has clear ideas and data and has made a credible control group. This is a high-level article, recommended to accept.

Author Response

Response to reviewers

We thank the four reviewers for their appreciation of our study and for the constructive comments. I would like to express my sincere gratitude to them for their invaluable comments and feedback on our scientific paper. Their insightful and constructive input significantly enhanced the quality and rigor of the research. Their thorough evaluation not only strengthened the clarity of the methodology and results but also improved the overall coherence of the paper. Their expertise and time devoted to the peer-review process have been instrumental in advancing the scientific discourse in our field. I am genuinely appreciative of their dedication to upholding scholarly standards. Their contributions have been instrumental in refining this work, and I extend my heartfelt thanks for their commitment to the advancement of science. You will find bellow a point-to-point answer at all the comments.

Reviewer 1

We thank the reviewer for his recommendation that our paper can be published to the scientific community.

Reviewer 2 Report

The Manuscript ijms-2648839 entitled “Multiple subtypes of GFAP-expressing glial cells in the dorsal root ganglion after nerve injury” by E.A. Konnova et al. It is a remarkable and interesting contribition to the characterization of satellite glial cells of dorsal root ganglia (DRG).

The study findings, with a variety of well-planned and performed techniques, clearly demonstrate the presence of glial fibrillary acidic protein in DRG satellite cells, which has long been questioned (Glial fibrillary acidic (GFA) protein in Schwann cells: fact or artifact? J Histochem Cytochem. 1982 Sep;30(9):912-8. doi: 10.1177/30.9.6182187). Another significant finding is the demonstration of some types of ion channels in satellite glial cells.

In the opinion of this reviewer, the work is of high quality, but before being accepted, the authors should consider including in the introduction the seminal works of Ennio Panese's group (Pannese E. The satellite cells of the sensory ganglia. Adv Anat Embryol Cell Biol. 1981;65:1-111. DOI: 10.1007/978-3-642-67750-2; Pannese E. The structure of the perineuronal sheath of satellite glial cells (SGCs) in sensory ganglia. Neuron Glia Biol. 2010 Feb;6(1):3-10. DOI: 10.1017/S1740925X10000037).

On the other hand, there is some confusion when establishing Schwann cell types. I recommend to the authors the work The Lamellar Cells of Vertebrate Meissner and Pacinian Corpuscles: Development, Characterization, and Functions. Front Neurosci. 2022 Mar 9;16:790130. DOI: 10.3389/fnins.2022.790130, which conducted an updated review of the ontogenetic differences of satellite cells and Schwann cells, myelinating and non-myelinating cells.

No comments

Author Response

Response to reviewers

We thank the four reviewers for their appreciation of our study and for the constructive comments. I would like to express my sincere gratitude to them for their invaluable comments and feedback on our scientific paper. Their insightful and constructive input significantly enhanced the quality and rigor of the research. Their thorough evaluation not only strengthened the clarity of the methodology and results but also improved the overall coherence of the paper. Their expertise and time devoted to the peer-review process have been instrumental in advancing the scientific discourse in our field. I am genuinely appreciative of their dedication to upholding scholarly standards. Their contributions have been instrumental in refining this work, and I extend my heartfelt thanks for their commitment to the advancement of science. You will find bellow a point-to-point answer at all the comments.

Reviewer 2

Comment 1.

In the opinion of this reviewer, the work is of high quality, but before being accepted, the authors should consider including in the introduction the seminal works of Ennio Panese's group(Pannese E. The satellite cells of the sensory ganglia. Adv Anat Embryol Cell Biol. 1981;65:1-111. DOI: 10.1007/978-3-642-67750-2; Pannese E. The structure of the perineuronal sheath of satellite glial cells (SGCs) in sensory ganglia. Neuron Glia Biol.2010 Feb;6(1):3-10. DOI: 10.1017/S1740925X10000037).

Response:

The introduction was enriched with the indicated paper and you can see the changes highlighted in the text:

''After nerve injury, SGCs increase coupling with neighbouring SGCs by upregulating connexin 43 (Cx43) gap junctions and form a network of SGCs that surround multiple neurons [16, 17]. Also after injury, SGCs may bridge separate perineuronal sheaths, forming complex sheath structures, which exhibit complex neuron-SGC boundaries with projections [18]''

Comment 2.

On the other hand, there is some confusion when establishing Schwann cell types. I recommend to the authors the work The Lamellar Cells of Vertebrate Meissner and Pacinian Corpuscles: Development, Characterization, and Functions. Front Neurosci.2022 Mar 9;16:790130. DOI: 10.3389/fnins.2022.790130, which conducted an updated review of the ontogenetic differences of satellite cells and Schwann cells, myelinating and non-myelinating cells.

Response:

In the introduction was included the following text, which is highlighted:

''Recent transcriptomic data identified subpopulations of SGCs within the DRG, sympathetic ganglion and trigeminal ganglion in naïve mice [8, 14]. There may be further diversity generated after nerve injury, as SGCs become activated and undergo expression changes [6]. In the peripheral terminals of Aβ-primary sensory neurons are present another type of non-myelinating Schwann-like cells which are called terminal glial cells that primarily work as low-threshold mechanoreceptors and potentially exhibit neurotransmission properties during action potential generation [15].''

Reviewer 3 Report

In the article: “Multiple subtypes of GFAP-expressing glial cells in the dorsal root ganglion after nerve injury.” by Elena A. Konnova et al, the authors characterize GFAP-expressing cells in DRG after spared nerve injury in hGFAP-CFP mouse line. Authors provide immunofluorescence results without/with FACS CFP+ FACS sorting, Western Blot and electrophysiology (whole cell patch clamp) suggesting that nerve injury evokes GFAP upregulation in two types of glial cells present in DRG: satellite glial cells and in non-myelinating Schwann cells. The scientific approach presented in the manuscript is interesting, and the results are original. My critical remarks mainly concern methodological elucidations and results presentation.

Comments:

  1. Please improve the title of your manuscript based on instructions for authors. The title should be “concise, specific and relevant”. In the current shape, based on manuscript content, the phrase “multiple subtypes” is confusing. I recommend also specifying the nerve injury model used in experiments.
  2. There is no indication either in the body text of the Article or in the section Institutional Review Board Statement permit date and permit number of the local Ethics Committee confirming that experiments on animals met necessary ethical criteria. Please complete.
  3. Instead of a summary of information about the aim, methods, and results presented in the manuscript, the Abstract, in its current shape is not an objective representation of the article. Please improve it based on instructions for authors.
  4. Please improve the usage of the abbreviations throughout the article. Once entered, the abbreviation should be used consistently. I found a lack of full names e.g. SNI (line 104). The list of abbreviations could be helpful.
  5. Results descriptions and presentations are not precise, hard to understand, and sometimes confusing. Especially Y-axis titles in graphs presented in Figs. should be improved. Please add verifiable units in the graphs presented in Figs 1-6. For units [%] please add a reference group and/or describe which values were considered as 100%. For “cells/mm2”- did you mean >number of cells/mm2<? What did you mean by “ROI” in the Y-axis title for Fig 5? Fig. 6 H, I – “normalized to total”, why not control protein? Please add rationale for different ways of Western Blot data calculations. 
  6. Please correct and complete the material and methods section to understand the causal connection of used methods, enable accurate assessment of the quality of performed studies, and redo future experiments. I found incomplete information e.g.: regarding SNI methodology (line 373); amount of protein taken for each well in West Blot; methodology of staining for cryosections, amount of DRG sections taken for immunofluorescent analysis, method of immunofluorescent, western blot data calculations.

Author Response

Response to reviewers

We thank the four reviewers for their appreciation of our study and for the constructive comments. I would like to express my sincere gratitude to them for their invaluable comments and feedback on our scientific paper. Their insightful and constructive input significantly enhanced the quality and rigor of the research. Their thorough evaluation not only strengthened the clarity of the methodology and results but also improved the overall coherence of the paper. Their expertise and time devoted to the peer-review process have been instrumental in advancing the scientific discourse in our field. I am genuinely appreciative of their dedication to upholding scholarly standards. Their contributions have been instrumental in refining this work, and I extend my heartfelt thanks for their commitment to the advancement of science. You will find bellow a point-to-point answer at all the comments.

Reviewer 3

Comment 1.

Please improve the title of your manuscript based on instructions for authors. The title should be “concise, specific and relevant”. In the current shape, based on manuscript content, the phrase “multiple subtypes” is confusing. I recommend also specifying the nerve injury model used in experiments.

Response:

The title was changed to avoid confusion and the nerve injury model was stated:

''Characterization of GFAP-expressing glial cells in the dorsal root ganglion after spared nerve injury.''

Comment 2. There is no indication either in the body text of the Article or in the section Institutional Review Board Statement permit date and permit number of the local Ethics Committee confirming that experiments on animals met necessary ethical criteria. Please complete.

Response:

The authorisation numbers were included in the ''Study approval'' and '' Institutional Review Board Statement'', please see the changes:

''Study approval

All experiments involving animals were approved by the Committee on Animal Experimentation of the Canton de Vaud, Switzerland, with the experiment licenses VD3068.2b and VD1339.10, in accordance with the Swiss Federal Laws on Animal Welfare and the guidelines of the International Association for the Study of Pain guidelines for the use of animal in research [56].

Institutional Review Board Statement: The animal study protocol was approved by the Committee on Animal Experimentation of the Canton de Vaud, Switzerland, with the experiment licenses VD3068.2b and VD1339.10, in accordance with the Swiss Federal Laws on Animal Welfare and the guidelines of the International Association for the Study of Pain guidelines for the use of animal in research [56].''

Comment 3. Instead of a summary of information about the aim, methods, and results presented in the manuscript, the Abstract, in its current shape is not an objective representation of the article. Please improve it based on instructions for authors.

Response:

The abstract was re-written based on instructions for authors and the changes are presented here:

''Abstract: Satellite glial cells (SGCs), enveloping primary sensory neurons' somas in the dorsal root ganglion (DRG), contribute to neuropathic pain upon nerve injury. Glial fibrillary acidic protein (GFAP) serves as an SGC activation marker, though its DRG satellite cell specificity is debated. We employed the hGFAP-CFP transgenic mouse line, designed for astrocyte studies, to explore its expression within the peripheral nervous system (PNS) after spared nerve injury (SNI). We used diverse immunostaining techniques, western blot analysis, and electrophysiology to evaluate GFAP+ cell changes. Post-SNI, GFAP+ cell numbers increased without proliferation. Similarly, the proportion of injured ATF+ neurons increased, as their association with CFP+ cells. GFAP+ FABP7+ SGCs increased, yet 75.5% of DRG GFAP+ cells lacked FABP7 expression. This suggests a significant subset of GFAP+ cells are non-myelinating Schwann cells (nmSC), indicated by their presence in the dorsal root but not the ventral root, rich in unmyelinated fibers. Additionally, patch clamp recordings from GFAP+ FABP7- cells lacked SGC-specific Kir4.1 currents, instead displaying outward Kv currents expressing Kv1.1 and Kv1.6 channels specific to nmSCs. In conclusion, this study demonstrates increased GFAP expression in two DRG glial cell subpopulations post-SNI: GFAP+ FABP7+ SGCs and GFAP+ FABP7- nmSCs, shedding light on GFAP's specificity as a SGC marker after SNI.''

Comment 4.

Please improve the usage of the abbreviations throughout the article. Once entered, the abbreviation should be used consistently. I found a lack of full names e.g. SNI (line 104).The list of abbreviations could be helpful.

Response:

A list of abbreviations have been made at the end of the study. Changes have been made to the text as follows:

''2.1. Increase in GFAP-expressing cells in the DRG after spared nerve injury (SNI) without proliferation.''

Comment 5.

Results descriptions and presentations are not precise, hard to understand, and sometimes confusing. Especially Y-axis titles in graphs presented in Figs. should be improved. Please add verifiable units in the graphs presented in Figs 1-6. For units [%] please add a reference group and/or describe which values were considered as 100%.

Response:

We added clearer indications in all figures with the units [%]. Therefore, for Figure 1C we have included both in the main text and also in figure legend, these clarifications:

''While, GFAP protein created the cytoskeletal structure of glial cells and found in the cell’s processes [33], while the endogenous CFP signal labels the cell body of non-neuronal cells in the DRG (Figure 1A-B). To clearly confirm that the GFAP+ labelled cells are CFP+, we sorted the cells from the DRG by FACS and immunostained them with the GFAP antibody to reveal that 78.4% ± 11.0 S.D., from the total of CFP+ cells, were co-labelled with the GFAP antibody (Figure 1C-D). We confirmed that none of the CFP+ cells are Iba1+ cells in the DRG (Figure 1E-F).

Figure 1: Validation of CFP expression in the DRG of hGFAP-CFP mice. Representative images of immunofluorescence staining with antibody against GFAP of contralateral (A) and ipsilateral (B) L4 DRG 7 days after SNI. Top panels show the anatomical structure of the DRG, followed by zoomed image of the endogenous CFP signal (green), antibody staining (red) and merge, with arrows   pointing to co-localisation. CFP+ FACS sorted cells after dissociation of ipsilateral L3-L5 DRG at 7 days after SNI were cultured overnight and immunostained with antibody against GFAP (red) to allow for clear quantification of co-labelling of CFP+ GFAP+ cells (green, N=6 DRG cultures) (C-D). Representative images of immunofluorescence staining with antibody against macrophage marker Iba1 of contralateral (E) and ipsilateral (F) L4 DRG 7 days after SNI, followed by zoomed image of the endogenous CFP signal (green), antibody staining (red) and merge, with arrowheads pointing to CFP+ Iba1- cells.''

For Figure 3D we have included in the main text this indication:

''The increase in CFP signal occurred in the L3 and L4 DRG, which contained about 21.5% ± 9.9 S.D. and 36.5% ± 7.8 S.D. of ATF3+ injured neurons, respectively (Figure 3A-D). The L5 DRG contained mainly uninjured neurons from the spared sural nerve, therefore it expressed a negligible amount of ATF3 in neurons after SNI (6.6% ± 4.5 S.D.). The percentage values of ATF3+ neurons were calculated based on NeuN staining. In addition, CFP+ cells were found more often close to ATF3+ injured neurons, rather than ATF3- uninjured neurons in the L3 and L4 DRG at 2, 4 and 7 days after SNI (Figure 3E).''

Comment 5. For “cells/mm2”- did you mean >number of cells/mm2<?

Response:

To be clear, changes have been made in the figures indicating the number of cells by the symbol #. Therefore, changes in figures were made in Figure 2D, Figure 2E, Figure 3E, Figure 4G, Figure 4H.

Comment 5. What did you mean by “ROI” in the Y-axis title for Fig 5?

Response:

Clarifications have been added in the figure legend, as follows:

''Figure 5: CFP is expressed in non-myelinating Schwann cells (nmSC) in the peripheral nervous system of hGFAP-CFP mice. (A) Representative images of immunofluorescence staining against GFAP in L4 DRG with ventral root (VR), dorsal root (DR) and spinal nerve (SpN) at 7 days after SNI. (B) Representative image of endogenous CFP+ signal in transverse section of the contralateral and ipsilateral sciatic nerve 7 days after SNI. (C) Quantification of the area of CFP+ signal over the entire region of interest (ROI) manually drawn around the perimeter of the transverse section of the nerve (N=4 mice, paired student’s t-test). (D) Representative images of immunofluorescence staining against myelinating Schwann cells marker myelin basic protein (MBP) at 7 days after SNI. Top panels show the anatomical structure of the DRG with the outlined region of interest containing neuronal somas, followed by zoomed image.''

Comment 5. Fig. 6 H, I – “normalized to total”, why not control protein? Please add rationale for different ways of Western Blot data calculations.

Response:

The sub-section of ''Western Blot'' was improved with more details. Whereas the details are not giving a pertinent rationale, why we used a normalization to total protein by Ponceau instead of a normalization to control protein by Tubuline, we will try to clarify in the comments. In Figure 2G, the western blot was done from all DRG tissue and the GFAP and Tubuline were indicated, whereas in Figure 6F-I the western blot was done from CFP+ cells sorted by FACS. The two approaches are different and we decided to for technical reasons to show the normalization against the total amount of protein or the control protein. Our comments in the ''Western blot'' sub-section are:

''Chemiluminescence was detected by using an imaging system (LAS-4000 Imaging System) coupled with an integrated CCD camera. Protein quantification was performed using ImageJ software (Fiji; Schindelin et al., 2012), analysing the ratio between the intensity of the protein band against the tubulin (for DRG tissue) or total amount of protein (for CFP+ cells sorted by FACS). The quantified signals for the proteins of interest were normalized to reference protein signals.''

Comment 6. Please correct and complete the material and methods section to understand the causal connection of used methods, enable accurate assessment of the quality of performed studies, and redo future experiments. I found incomplete information e.g.: regarding SNI methodology (line373);

Response:

Throughout the ''Materials and Methods'' sections improved clarification have been made, along with minor changes into figure legends. Regarding the SNI methodology, the following comments have been added in the sub-section of ''Animals'':

''Animals

Experiments were performed on homozygote TgN(hGFAP-ECFP)-GCED referred to as hGFAP-CFP mice (a generous gift from Paola Bezzi, Department of Fundamental Neurosciences, University of Lausanne) adult male and female mice [32]. The animals were housed in standard cages with free access to food and water at 22 ± 0.5°C under a controlled 12 hr light/dark cycle. The SNI or sham surgeries were performed unilaterally as described previously [56, 57]. Briefly, mice were anesthetized with isoflurane (Piramal). The trifurcation of the sciatic nerve was exposed. The two branches of the nerve (tibial and common peroneal) were ligated with silk 6.0 suture and sectioned, while the sural branch remained spared. Muscles and skin were sutured and animals were left to re-cover before the transfer into standard cages. Tissue was collected after terminal intraperitoneal injection of pentobarbital (50 mg/kg).''

Comment 6. Amount of protein taken for each well in West Blot; western blot data calculations.

Response:

In the ''Western Blot'' subsection we have added the amount of protein used and the data calculations, as indicated in the highlighted text:

''Western Blot

Western blot were performed on DRG tissue or CFP+ cells sorted by FACS. The samples were lysed in 100 mM Tris HCl pH 6.8, 2% SDS, 10% glycerol, Complete Protease inhibitor cocktail tablets (Roche). Soluble fractions were recovered in supernatants after 20 min of centrifugation at 10 000 rpm at 4 °C. Protein concentration was measured using Bradford BSA standard scale (Sigma-Aldrich) with Assay dye (BioRad) and loaded at an amount of 40 µg/well. For western blot of GFAP protein in L3, L4, L5 DRGs tissue the normalisation was done to tubulin, whereas the quantification of Kv1.1 and Kv1.6 protein level in CFP+ cells sorted by FACS was normalised to total protein level.''

Comment 6. Methodology of staining for cryosections, amount of DRG sections taken for immunofluorescent analysis, method of immunofluorescent

Response:

The sub-section of '' Immunofluorescence'' was improved and the following comments have been made in the text and figure legends to allow a better clarification about the experimental steps. Even more, the ''Supplemental table 2: List of statistical test for significance for each figure'' have been improved and the changes are highlighted in yellow. The changes in the text can be found also here:

''Immunofluorescence

Mice were terminally anesthetized by i.p injection of pentobarbital and transcardially perfused with saline, followed by 4% paraformaldehyde (PFA, Sigma-Aldrich) in PBS. L3-L5 DRGs were dissected and post-fixed at 4°C for 90 min and then transferred in 20% sucrose in PBS for 24h. The DRG were rapidly frozen in embedding solution (Tissue-Tek O.C.T. Compound) to be cut in 12 μm thickness section with a cryostat, directly onto slides for immunostaining. Alternatively, immunostainings were done on dissociated in vitro culture after overnight incubation on coverslips that were previously coated with poly-D-lysine. The coverslips were incubated with 4% PFA in PBS for 2h to fix the cells. Following PBS rinsing, the coverslips or cryosections were immersed in a blocking solution, comprising 10% normal goat serum (NGS, Vector Laboratories) or normal horse serum (NHS, Vector Laboratories), and 0.3% Triton X-100 from Sigma-Aldrich in PBS, for 30 minutes at room temperature.

Epifluorescence images of the DRG sections were acquired at x20 magnification with Zeiss Axio Scan.Z1 slide scanner using the Zeiss Blue software, or at x20 or x63 with AxioVision (Zeiss). Fluorescence intensity and exposure time were kept constant for all images. Cell counting were performed using the Cell Counter Plugin in ImageJ software (Fiji).''

Reviewer 4 Report

Dear Authors,

I congratulate you on your complex study regarding cell lines in peripheral nerves.

However, there are some aspects that require your attention.

You need to insert a conclusions section.

Insert a limitations section of your study.

In the discussion section underline the need for future  development and study of the aspects  on human subjects and translation of these findings into clinical practice.

One field that will benefit from the present discoveries will be oncology, one reference could be the article by Vrînceanu D, Dumitru M, Åžtefan AA, Mogoantă CA, Sajin M. Giant pleomorphic sarcoma of the tongue base - a cured clinical case report and literature review. Rom J Morphol Embryol. 2020 Oct-Dec;61(4):1323-1327. doi: 10.47162/RJME.61.4.34. PMID: 34171081; PMCID: PMC8343483.

Insert a list of abbreviations at the end of the manuscript.

Looking forward to see the improved final version of your article.

Author Response

Response to reviewers

We thank the four reviewers for their appreciation of our study and for the constructive comments. I would like to express my sincere gratitude to them for their invaluable comments and feedback on our scientific paper. Their insightful and constructive input significantly enhanced the quality and rigor of the research. Their thorough evaluation not only strengthened the clarity of the methodology and results but also improved the overall coherence of the paper. Their expertise and time devoted to the peer-review process have been instrumental in advancing the scientific discourse in our field. I am genuinely appreciative of their dedication to upholding scholarly standards. Their contributions have been instrumental in refining this work, and I extend my heartfelt thanks for their commitment to the advancement of science. You will find bellow a point-to-point answer at all the comments.

Reviewer 4

Comment 1.

You need to insert a conclusions section.

Response:

The ''Conclusions'' section has been included in the study:

''4. Conclusions

This study indicates two distinct GFAP+ cell subpopulations within the DRG using the hGFAP-CFP mouse model: FABP7+ SGCs and FABP7- nmSCs. While FABP7 has been preferred as marker for SGCs, our findings reveal that GFAP is not a comprehensive SGC marker. Approximately 24.5% of GFAP+ glial cells were identified as FABP7+ SGCs in the soma-rich DRG region. However, a notable fraction of FABP7+ SGCs did not express GFAP, even after nerve injury, indicating GFAP's inadequacy in labelling all SGCs. Moreover, 75.5% of GFAP-expressing glial cells in the soma-rich DRG lacked FABP7 and were thought to be nmSCs. Electrophysiological characterization confirmed the presence of GFAP+ nmSCs, displaying Kir4.1 currents characteristic of SGCs, large Kv outward currents, and expression of Kv1.1 and Kv1.6 channels typical of nmSCs. Although the study did not detect significant changes in Kv outward currents, the presence of GFAP+ nmSCs may still impact primary sensory neuron repair and sensitization following nerve injury. This insight will facilitate further investigations into the distinct roles of glial cell subpopulations in the DRG post-peripheral nerve injury.''

Comment 2.

Insert a limitations section of your study.

Response:

Certain limitations of the study have been amended as follows:

''Several limitations warrant consideration in this study. While our research identified distinct GFAP+ cell subpopulations within the DRG, it primarily relied on the hGFAP-CFP mouse model, potentially limiting its generalizability to other species or models. The use of FABP7 as marker for SGC raises questions about its universal applicability, especially since not all FABP7+ SGCs expressed GFAP, suggesting that GFAP may not be all-encompassing for SGC labelling. Additionally, the study's focus on the soma-rich region of the murine DRG may have overlooked potential variations in other regions. The lack of a definitive marker for nmSCs highlights the ongoing challenge in distinguishing these cells from other DRG constituents. Finally, while electrophysiological characterization offered valuable insights, it did not reveal significant functional changes in nmSCs post-injury, necessitating further investigation. Clarifying the specific roles of diverse glial cell subpopulations within the DRG remains a complex, evolving endeavour. ''

Comment 3.

In the discussion section, underline the need for future development and study of the aspects on human subjects and translation of these findings into clinical practice.

Response:

Our findings might have some aspects important for clinical practice and have been addressed in the ''Discussions'' section in the following paragraph:

''The study's findings have translational implications for understanding the diverse glial cell subpopulations within the DRG in the context of peripheral nerve injury. The identification of distinct subtypes, such as FABP7+ SGCs and FABP7- nmSCs, underscores the complexity of the DRG microenvironment. These insights may lead to the development of targeted therapies for neuropathic pain management by focusing on specific glial cell subtypes. However, challenges in accurately distinguishing these subpopulations due to the lack of definitive markers underscore the need for further research.''

Comment 4.

One field that will benefit from the present discoveries will be oncology, one reference could be the article by Vrînceanu D,Dumitru M, Åžtefan AA, Mogoantă CA, Sajin M. Giant pleomorphic sarcoma of the tongue base - a cured clinical case report and literature review. Rom J Morphol Embryol. 2020 Oct-Dec;61(4):1323-1327. doi: 10.47162/RJME.61.4.34. PMID:34171081; PMCID: PMC8343483.

Response:

The scientific article entitled ''Giant pleomorphic sarcoma of the tongue base - a cured clinical case report and literature review. Daniela Vrînceanu 1, Mihai Dumitru, AngheluÅŸ Adrian Åžtefan, Carmen Aurelia Mogoantă, Maria Sajin. Rom J Morphol Embryol. 2020 Oct-Dec;61(4):1323-1327. PMID: 34171081 PMCID: PMC8343483 DOI: 10.47162/RJME.61.4.34'' has taken our full attention. We have read it with great interest and followed the patient evolution through radiation therapy, oncological treatment and surgery consisting of partial glossectomy for the residual tumor.

Although, we find the study interesting for the scientific community, we appreciate that our own study investigating GFAP-expressing glial cells in the dorsal root ganglion after spared nerve injury in mice is treating a different subject that makes it hard to reference the above mentioned study. Still, we thank the reviewer for the outstanding reference mention.

Comment 5.

Insert a list of abbreviations at the end of the manuscript.

Response:

A list of abbreviations have been made at the end of the study.

Round 2

Reviewer 3 Report

The authors addressed all my critical remarks and applied necessary corrections to the current version of the manuscript.

The current version, in my opinion is ready for publication and will be interesting for the scientific community.